# Bioprinting Technologies and Bioinks for Vascular Model Establishment

**DOI:** 10.3390/ijms24010891

**Published:** 2023-01-03

**Authors:** Zhiyuan Kong, Xiaohong Wang

**Affiliations:** 1Center of 3D Printing & Organ Manufacturing, School of Intelligent Medicine, China Medical University (CMU), No. 77 Puhe Road, Shenyang North New Area, Shenyang 110122, China; 2Key Laboratory for Advanced Materials Processing Technology, Ministry of Education & Center of Organ Manufacturing, Department of Mechanical Engineering, Tsinghua University, Beijing 100084, China

**Keywords:** 3D bioprinting, organ manufacturing, tissue engineering, biomaterials, vascular networks

## Abstract

Clinically, large diameter artery defects (diameter larger than 6 mm) can be substituted by unbiodegradable polymers, such as polytetrafluoroethylene. There are many problems in the construction of small diameter blood vessels (diameter between 1 and 3 mm) and microvessels (diameter less than 1 mm), especially in the establishment of complex vascular models with multi-scale branched networks. Throughout history, the vascularization strategies have been divided into three major groups, including self-generated capillaries from implantation, pre-constructed vascular channels, and three-dimensional (3D) printed cell-laden hydrogels. The first group is based on the spontaneous angiogenesis behaviour of cells in the host tissues, which also lays the foundation of capillary angiogenesis in tissue engineering scaffolds. The second group is to vascularize the polymeric vessels (or scaffolds) with endothelial cells. It is hoped that the pre-constructed vessels can be connected with the vascular networks of host tissues with rapid blood perfusion. With the development of bioprinting technologies, various fabrication methods have been achieved to build hierarchical vascular networks with high-precision 3D control. In this review, the latest advances in 3D bioprinting of vascularized tissues/organs are discussed, including new printing techniques and researches on bioinks for promoting angiogenesis, especially coaxial printing, freeform reversible embedded in suspended hydrogel printing, and acoustic assisted printing technologies, and freeform reversible embedded in suspended hydrogel (flash) technology.

## 1. Introduction

Bioartificial blood vessel manufacture has been a long-term dream for human beings [1]. To date, engineered tissues with thickness greater than 1 mm are known to have difficulty in maintaining their normal cell viability without vascularization because the nutrient/oxygen diffusion limit is approximately 100–200 μm. The in vitro culture of high-density cells requires sufficient nutrients and oxygen to produce tissues with adequate vascularized structures [2,3,4,5].

Anatomically, blood vessels in human bodies are organized into tree-like networks with subsequent branches, which allow for the most efficient exchange of oxygen, nutrients and metabolic wastes (Figure 1). The tree-like arterial network contains aortas (diameter > 6 mm), arteries (diameter 1–6 mm), arterioles (diameter 0.1–1 mm) and capillaries (diameter 10–15 µm). Similarly, the tree-like venous network contains venae (diameter 5–20 mm), veins (diameter 1–5 mm), and venules (diameter ~ 200 µm). Large aortas deliver oxygenated blood from the heart to arteries, then to arterioles, and finally to the capillary bed. On the contrary, venules drain the oxygen-depleted blood from the capillaries into venules, then into veins and finally venae [6]. Similarly, in a large vascularized organ, both the arterial and venous networks are necessary for the internal blood circulation. So, in order to achieve the internal blood circulation in a bioartificial organ, the basic vascular elements should include not only the hierarchical arterial and venous trees, but also the medial capillaries in between the arterial and venous trees.

Over the last three decades, the main strategies to promote angiogenesis include the ingrowth of newly formed capillaries within the implanted structures from the host, and by stimulating cell seeding with the help of prevascularized structures to achieve rapid blood supply within the engineered structures [6,7,8]. After implantation, prevascularized channels within the construct interconnect with the adjacent vascular network for rapid perfusion [9,10,11,12]. In addition, the cell-based techniques for promoting angiogenesis also involve the co-culture of stem cells with endothelial cells (ECs), which can provide functional vasculature through constructs linked with host vasculature. Several groups have reported that ECs formed new blood vessels when they are co-cultured with neural progenitor cells [13] and mesenchymal stem cells (MSCs) [14]. For example, Koike et al. seeded human umbilical vein ECs (HUVECs) and 10T1/2 mesenchymal precursor cells in three-dimensional (3D) fibronectin type I collagen gel and implanted the 3D constructs into mice. A network of long-lasting blood vessels was constructed in mice and stayed stable in vivo for one year after operation [15]. Nevertheless, it was gradually realized that there were many inherent limitations for the traditional solid-scaffold-based ‘top-down’ tissue engineering approaches to build thick vascular tissues and organs.

In parallel, with the rapid development of 3D printing technologies, automatic vascular building approaches have made great contributions to vascular tissue/organ construction and angiogenesis. These technologies involve the printing of cell-laden hydrogels and polymeric scaffolds. By directional manipulation of different cell types in a 3D space, a hierarchical vascular network with multi-scale and multi-branched structures, can be built in a layer-by-layer way, simulating the heterogeneity of natural vascular tissues [16,17].

Figure 2 summarizes the methods of traditional tissue engineering and 3D bioprinting approaches to construct vascular tissues. There are many advantages for vascular tissue/organ engineering with the 3D bioprinting technologies. Especially, it is convenient to print different cell types and polymer systems using combined multi-nozzle extrusion-based 3D bioprinters. With the advances of polymeric bioinks and stem cells, it is now possible to produce various blood vessels ranging from micrometers to millimeters in size with micron precision [18]. This has greatly prompted researchers to explore new printing techniques to solve the complex vascular network construction problems.

## 2. 3D Bioprinting Technology for Building Blood Vessels

In this section, we mainly discuss the 3D bioprinting technologies in vascular model construction, including the printing methods and their applications. The traditional bioprinting technologies, such as inkjet printing, laser-assisted printing, and extrusion printing, still have unique advantages in vascular model construction and are constantly developing. Several new printing technologies, with advantages in biocompatibility, printing resolution and vascular complexity, have also been discussed.

### 2.1. Inkjet Bioprinting

Inkjet bioprinting is firstly introduced by revising the commercial two-dimensional (2D) printing equipment (Figure 3). A solution of hydrogel prepolymer with encapsulated cells, called bioink, is stored in the cartridge. The ink cartridges are then attached to the printer head and act as bioink sources during the electronically controlled printing process. During printing, the printhead is deformed and squeezed by thermal or piezoelectric actuators to produce droplets of controllable size. Among them, the working frequency of piezoelectric inkjet printers reaches 15–25 kHz, which will cause certain ultrasonic damage to cells [19]. While in thermal inkjet printing, cells are heated for 2 μs, and the damage is very low. The viability of ink-printed cells was around 90% [20]. Compared with piezoelectric inkjet printing, thermal inkjet printing has been more widely used in tissue engineering and regenerative medicine.

In 2009, Cui et al. used a modified thermal inkjet printer to print cells with a quasi-3D structure. The bioink was composed of human microvascular ECs (HMVECs) and fibrin [21]. After 21 days of in vitro culture, the printed HMVECs aligned in the thin grid fibrin surface and proliferated to form a confluent lining (Figure 3b). The thin grid structure was further revealed using confocal laser scanning images. It was hoped that the HMVECs on the thin grid construct could form true 3D blood vessels with the improvement of their inkjet printer.

Electrohydrodynamic (EHD) inkjet bioprinting is a unique material injection technology that is related to traditional drop-on-demand inkjet printing, but produces a much smaller droplet volume [22,23]. Unlike traditional drop-on-demand inkjet printing, EHD inkjet printing uses an electric field rather than thermal or acoustic energy to create the fluid flow, which is needed to deliver the ink to the substrate [24]. By applying an electric potential difference between the printhead nozzle and the substrate, the ink fluid is charged, and the combination of electrostatic and capillary forces leads to the formation of a characteristic conical shape, the Taylor cone. With the use of a pulsed potential, individual drops can be ejected driven by the electric force [25]. This drop formation mechanism allows EHD inkjet printing to achieve droplets up to 10^−15^ cm^3^ in volume and a consequent line resolution of about 100 nm to a few microns (dependent on ink surface contact angle) after printing on substrates. This improves the potential feature resolution compared to that of extrusion-based or conventional inkjet 3D printing by about three orders of magnitude.

**Figure 3 ijms-24-00891-f003:**
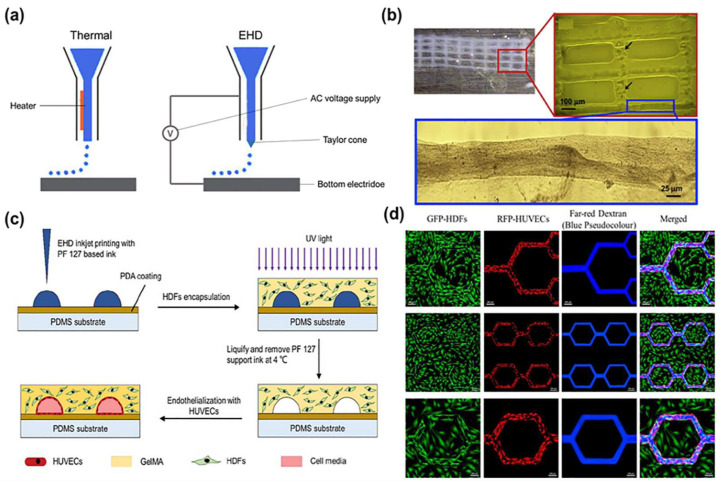
(**a**) Schematic diagrams of the thermal inkjet printing and EHD inkjet printing techniques. (**b**) Schematic of the manufacturing process using a sacrificial Pluronic F127 solution to form the microvasculature followed by casting a cell-containing methacrylated gelatin (GelMA) suspension and photopolymerization. Adapted from Ref. [21]. (**c**) Printed fibrin scaffold using modified thermal inkjet printer. Only minor deformations of the printed pattern were noticed at y-axis. (**d**) Perfusion of fluorescent dextran solution into a GFP-HDFs/RFP-HUVECs co-culture construct. Far-red dextran flow is marked as blue pseudocolour after image processing, reported by Zheng et al. (**c**,**d**) adapted from Ref. [26].

In 2020, Zheng et al. employed an EHD inkjet printing-based approach using variable materials to create cell-embedded microvascular structures with significantly higher spatial resolution than extrusion-based bioprinting [26]. Specifically, Pluronic F127 and gelatin methacryloyl were used as sacrificial templates and permanent matrices, respectively. Human dermal fibroblasts and umbilical vein ECs were successfully co-cultured within the engineered microvascular constructs to form a supportive extracellular matrix (ECM) and a functional endothelial layer (Figure 3c). It enabled the fabrication of microvascular structures with feature sizes down to 30 μm, mimicking capillaries (Figure 3d). This opens up new avenues for studying complex vascular models, particularly from arterioles to venules, in native vasculatures.

### 2.2. Laser-Assisted Bioprinting

Laser-assisted bioprinting (LAB) originated from laser direct writing [27,28,29] and laser-induced transfer (FIFT) technologies (Figure 4) [30,31]. A key part of a LAB system is that the donor layer responds to laser stimulation. The donor layer consists of a “ribbon” structure containing an energy-absorbing layer (such as titanium or gold) on top and a bioink solution on the bottom. During printing, focused laser pulses are applied to stimulate a small area of the absorber layer. This laser pulse vaporizes a portion of the donor layer, creating high-pressure bubbles at the interface of the bioink layer and propelling the suspended bioink. The falling bioink droplets are collected on the receiving substrate and subsequently crosslinked [32,33]. This method was used to print spots of human adipose stem cells (HASCs) and endothelial colony-forming cells, with the result of a blood vessel-like network formation [34]. Based on laser-induced forward transport, Koch et al. directly printed capillaries with diameters of tens of microns in 2021. When the ECs were printing on Matrigel (Figure 4b), nearly all of them could form tubular-like vessels with a lumen inside [35].

Compared to other printing methods, LAB avoids direct contact between the dispenser and bioink, and does not suffer from head clogging, limited cell density, low resolution and other issues [36]. Cells would not be damaged by shear stresses. For example, Kerouredan et al. used LAB to pattern collagen I hydrogels, and co-cultured ECs and MSCs on the pattern. After 3D printing, cells could cover the collagen pattern with capillary-like structures [37]. This demonstrated that LAB can pre-organize ECs into high cell densities, creating well-defined vascular networks in the engineered constructs.

The main limitation of the LAB techniques in vascular construction is the need to maintain a thin and uniform bioink film (5–20 μm) on the donor layer. This step is technically challenging. In 2020, Orimi et al. proposed a new LAB method called laser-induced side transfer (LIST) [38]. This method uses low energy nanosecond laser (wavelength: 532 nm) pulses to generate a transient microbubble at the distal end of a glass microcapillary supplied with bioink. Microbubble expansion results in the ejection of a cell-containing micro-jet perpendicular to the irradiation axis. With this technique, the team printed HUVECs-containing drops and indicated that the LIST-printed cells could preserve their angiogenic junctional phenotypes. Subsequently in 2022, this team demonstrated that endothelial tubulogenesis could be spatially controlled by LIST bioprinting a HUVECs–laden bioink containing fibrinogen [39]. Though the LIST, with a spatial resolution of 165 μm for the tested conditions, is lower than the one (10–140 µm) attained by traditional LAB techniques for similar cell types, it is more efficient for vascular model printing than traditional LAB techniques.

In 2017, Keriquel et al. demonstrated the capability of LAB in vivo. A major advantage derived from LAB is the short printing time (only a few seconds), which can be easily translocated to in vivo experiments [40]. Similarly, Kerouredan et al. in-situ printed ECs into mouse calvarial bone defects prefilled with collagen containing MSCs and vascular endothelial growth factors (VEGFs) using this LAB technology in 2019 [41]. ECs were printed into ring, disc, and a crossed circle in vivo. In-situ printing of HUVECs was shown to significantly increase in vivo vascularization and promote bone regeneration compared to random seeding. So, LAB has provided a new perspective for clinical applications.

### 2.3. Extrusion Bioprinting

Extrusion printing is currently one of the most frequently used methods for promoting angiogenesis and vasculature formation (Figure 5). It is also the most widely used bioprinting technology for complex vascular model establishment due to its ease of use, fast prototyping, low cost, and compatibility with various hydrogels in which cells can be encapsulated [42,43]. During 3D printing, the cell-laden hydrogel bioink is generally dispensed through a nozzle or syringe to form filamentous fibers. After extrusion, the cell-laden hydrogel bioink, in the form of cells encapsulated in cylindrical filaments, is deposited layer-by-layer in a precise manner into targeted 3D custom-shaped structures [44]. Physical or chemical crosslinking that occurs after extrusion enables rapid curing of the bioink to ensure the geometric fidelity of the bioprinted structures. To date, extrusion-based 3D bioprinting has been adapted by many research groups to create blood-vessel-like networks [45,46].

The first extrusion-based 3D bioprinting was reported in 2005 by Prof. Wang et al. at Tsinghua University [47,48]. It is also the first true 3D bioprinting technology using rapid prototyping (RP), i.e., the discrete-stacking principle, to produce large scale-up living tissues. Cell-laden gelatin-based hydrogels, such as gelatin/alginate, gelatin/chitosan, gelatin/algainte/fibrinogen, gelatin/hyaluronate, containing hepatocytes or adipose-derived stem cells (ASCs), were printed into large grid (or mesh) structures with nutrient and metabolite exchange channels based on a one nozzle 3D bioprinter [47,48]. After this report, a series of two or multi-nozzle 3D printers were created for automatic organ manufacturing. For example, a hybrid hierarchical polyurethane-cell/hydrogel construct was rapidly manufactured using a double-nozzle extrusion bioprinter (Figure 5b–f). An oval bioartificial liver precursor was constructed with perfusable branching vessels and ASCs encapsulated inside the gelatin-based hydrogels formed capillaries in between the branched vessels [49,50].

**Figure 5 ijms-24-00891-f005:**
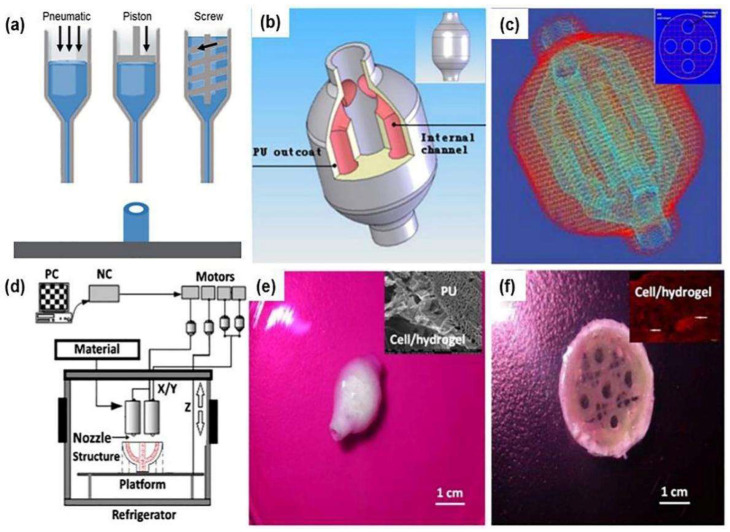
(**a**) Extrusion bioprinting nozzle (or syringe) systems (pneumatic, piston and screw). (**b**) A digital computer-aided design (CAD) model with an outlook and an internal branched network of the vascular networks. (**c**) A common layer interface (CLI) file. (**d**) Schematic illustration of the fabrication process of a layered vessel model using a dual-jet extrusion printer by Wang et al. (**e**) A 3D printed hybrid hierarchical polyurethane-cell/hydrogel structure. (**f**) The middle part of (**e**) with branched/grid internal cell/hydrogel channels. (**b**–**f**) are adapted with permission from Ref. [49]. Copyright © 2023 Elsevier B.V. All rights reserved.

With these inventions, Prof. Wang has solved all the bottleneck problems encountered by tissue engineers, biomaterial researchers, pharmacists and other scientists for more than several decades. These bottleneck problems include hierarchical vascular/nerve network construction with a fully endothelialized inner surface, step-by-step ASC differentiation in a 3D construct, energy system modeling, and high-throughput drug screening [51,52,53,54,55,56,57,58,59,60].

Afterwards, a great deal of similar printing methods were reported by many other groups, such as parallel printing of blood vessel tissues with multiple bioinks [61,62,63]. For instance, De Moor et al. generated prevascularized spheroids by combining ECs with fibroblasts and ASCs as supporting cells in 2021 [64]. Small diameter capillary-like microvessels were formed in 3D printed scaffolds. In the same year, Liu et al. printed centimeter-scale tissue constructs by endothelializing the printed macroscale channels, with capillary networks through cell self-assembly [65]. In 2022, Liu et al. developed centimeter-scale cardiac patches with well-organized microvasculature via the same extrusion bioprinting technology [66]. It has proven that it is super for vascular tissue formation with two vascular cell lines rather than only one cell line.

### 2.4. Coaxial Bioprinting

Coaxial bioprinting is suitable for continuous fabrication of tubular structures (Figure 6). The key feature of coaxial bioprinting is the two-layer nozzle, which enables the co-extrusion of two different bioink formulations in a core-shell manner to form a circumferentially layered tubular structure [67,68,69,70,71,72]. Mechanical properties of the tubular structures, such as toughness, stretchability, and compression resistance, can be well adjusted by selecting appropriate hydrogel materials. By adjusting the inner and outer diameters of the coaxial nozzles and printing parameters, small tubular structures with a small diameter (average of several hundred microns) can be produced. Thus, it is more suitable for small arterial/venous vascular model construction in bioartificial organs.

In 2015, Zhang et al. from the University of Iowa demonstrated the fabrication of vascular tissue using a coaxial extrusion-printing technique. An alginate hydrogel loaded with human umbilical vein smooth muscle cells (HUVSMCs) was printed. The perfusion efficacy of the 3D construct was certified (Figure 6b) [67]. Long-term culture of the 3D printed construct demonstrated that smooth muscle matrix and collagen were deposited around the peripheral and luminal surfaces. Analogously, Millik et al. fabricated unmodified Pluronic F-127 (F127) and F127-diurethane methacrylate (F127-BUM) hydrogels into tubular structures using the coextrusion technique in 2019 [68]. By varying the printing parameters, tubes or coaxial filaments with different cross-sectional geometries were generated (Figure 6c).

In 2020, Liang et al. prepared a novel hydrogel with high-strength by inserting N-acryloylglycinamide (NAGA) monomer with intramolecular multiple hydrogen bonds into the macromolecular chain of methacrylated gelatin (GelMA) hydrogel. Through introducing NAGA monomers and nanoclay (Clay) into the GelMA hydrogel, a small-diameter vascular stent was produced using the coaxial bioprinting technique [69]. A high-performance vascular stent with a tensile strength of 22 MPa, a stretchability of 500%, a Young’s modulus of 21 MPa, an anti-fatigue performance of 200 cycles, and a suture retention strength of 280 gf was obtained. After adding human umbilical vein ECs at the same time, good biocompatibility was achieved.

Through coaxial printing, it is easy to obtain vascular networks containing an endothelial layer. Nevertheless, most of the natural vascular networks contain multilayered tissues, in addition to the innermost endothelial layer. Especially, there are also middle smooth muscle layer and outermost fibroblast layer for large blood vessels. It is rational to develop new 3D bioprinters with multi-layer coaxial printing (Figure 6d) [70,71,72,73]. In 2020, Bosch et al. used sodium alginate and collagen hydrogels as bioinks, pluronics as sacrificial layers for three-layer coaxial printing. During printing, the inner layer hydrogel was loaded with HUVECs, and the outer layer hydrogel was loaded with human aortic smooth muscle cells (HASMCs), a bilayer blood vessel was built, mimicking the natural vascular structures (Figure 6e) [72]. Realizing the high concentration of alginate molecules limited the cell behaviours, including adhesion, proliferation and initial stretching, the authors optimized the bioink with high concentrations of collagen to mimic the ECMs of natural tissues in 2022 [73].

Similar to other 3D bioprinting technologies, the mechanical properties of the coaxial 3D printed constructs rely on the rapid crosslinking of polymer molecules in the cell-laden hydrogels. Most of the natural polymeric hydrogels have limited mechanical strengths, which are not suitable for anti-sutural vascular model establishment. Meanwhile, the lumen size is generally in the hundreds of microns by coaxial printing. For clinical application, the correct anastomosis with internal blood vessels and long-term functional are always the key issues to be considered. In 2022, Wang et al. used a double-network hydrogel for microfluidic bioprinting. They adopted two distinct approaches to fabricate venous and arterial vessels closely resembling their native counterparts [74]. The vessels were connected with a mouse vena cava and demonstrated the potential of the bioprinted vascular conduits as vascular grafts in the future. The vessels were further infected with pseudo-typed severe acute respiratory syndrome coronavirus 2 (SARS-CoV-2) viral particles (pCoV-VPs) and subjected to antiviral drugs, illustrating the applicability of these vessels for in vitro vascular disease studying and drug testing.

In some respects, coaxial printing technology is considered the most effective way for obtaining small-diameter blood vessels. However, due to the limitation of the coaxial nozzle, the size of the printed tubular stent is restricted. A continuous tree-like bifurcated structure cannot be manufactured. There is a need for a more effective technology to solve the above problems.

### 2.5. Freeform Reversible Embedding of Suspended Hydrogel (FRESH) Printing

FRESH printing was proposed by Hinton et al. in 2015 [75]. Instead of printing cell-laden hydrogels in air, the extruded filaments were deposited in a supportive bath. In the supportive bath, gelatin microparticles act as a Bingham plastic during the printing process, behaving as a rigid body at low shear stresses but flowing as a viscous fluid at higher shear stresses. This means that there is little mechanical resistance as the needle-like nozzle passes through the bath, but the hydrogel could be extruded from the nozzle and deposited in right place. After 3D printing, the gelatin was removed at 37 °C, resulting in scaffolds with high resolution and excellent cell-friendliness (Figure 7a). Using this technique, Hinton and colleagues constructed a vascular model of human right coronary arterial tree around a chicken embryonic heart with internal trabeculae. It is another approach to predefine the complex vasculature structures through extrusion printing.

More recently, some other teams have also used this technique to print vascular tissues. For example, Lee et al. printed human heart components at various scales, from capillaries to whole organs in 2019 [76]. The obtained 3D constructs include neonatal capillaries of less than 10 μm, complex vasculatures, ventricular and tri-leaflet heart valves, and neonatal-scale human hearts (Figure 7b). The resolution of the FRESH printing can be increased in the supportive bath using a coacervation approach, resulting in a vasculature with denser networks and smaller diameters, including perfusable vessels around 100 μm in diameter.

In 2021, Kreimendahl et al. used fibrin and hyaluronic acid as single or hybrid bioinks for FRESH bioprinting [81]. It is important to know that due to the low viscosity of the printing materials, such as fibrin and hyaluronic acid, there has always been a lack of fidelity in complex vascular bioprinting with this technique. The best values for vascularization were achieved with a bioink composed of 1.0% fibrin and 0.5% hyaluronic acid. Compared with other tested bioinks, the vascular structure area and length were three times higher, and the structure volume and branch number showed almost four times higher values [81]. Later in 2022, Daniel et al. printed collagen vascular scaffolds via the FRESH technique to obtain a layered and branched vascular network, and proved the application potential of the scaffolds in the study of vascular diseases [82].

### 2.6. Acoustics-Assisted Bioprinting

Acoustic-assisted bioprinting is a new technology that generates pressure gradients in bioinks through acoustic vibrations and obtains pre-designed patterns with resolutions ranging from several micrometers to several hundred microns (Figure 7c) [77,83,84,85]. Different from traditional printing methods, acoustic-assisted bioprinting enables cells to migrate and aggregate in a non-contact state, and achieves highly ordered arrangements. Even the cell–cell distance can be regulated with little damage to cells, which is crucial for the construction of complex vascular structures.

Kang et al. used pressure fields formed by standing surface acoustic waves (SSAWs) to pattern ASCs and ECs in a catechol-conjugated hyaluronic acid (HA-CA) hydrogel to recapitulate the 3D collateral vascular architecture of mouse hindlimb muscles. The SSAWs-assisted scaffolds formed perfusable and aligned blood vessels in mice and demonstrated therapeutic angiogenesis in a mouse model with hindlimb ischemia [86]. In 2020, Petta et al. reported a new printing method called sound-induced morphogenesis (SIM) [78]. Different from the ultrasonic standing waves used in the previous sound-assisted printing, they used Faraday waves to achieve patterning (Figure 7d), with improved flexibility and freedom of printing. Meanwhile, the influence of ultrasonic standing waves on the printing substrate was overcome. More importantly, this approach employed a low initial density of cells, and cells with a predetermined local density finally self-assembled into functional multiscale vascular networks. It is believed that with further modification of this technique, the acoustic-assisted bioprinting can organize more valuable vascular networks with the pre-patterned cells.

### 2.7. Stereolithography (SLA) Bioprinting

In the construction of complex organ models, SLA is a 3D printing technique with some advantages (Figure 7e). Similar to LAB, SLA bioprinters use light to selectively solidify a bioink additively to build objects. Compared to other printing technologies that require point-by-point or line-by-line scanning to create an individual layer, SLA could drastically reduce the printing time by enabling the projection of an entire 2D design plane at once [87,88]. The printer only needs a movable platform in the z-axis direction, improving the printing efficiency with a sufficiently high spatial resolution. SLA has been used to fabricate a variety of complex cell-laden structures including vascular networks [79,89,90,91,92,93,94].

In 2020, Alexander et al. used projection-based, multi-material stereolithographic bioprinter and an enzymatically degradable sacrificial photoink to print vascular models containing perfusable, endothelial cell-lined channels that remained stable for 28 days in culture [91]. Before this report, Grigoryan et al. used SLA printing to model an interconnected 3D structure of alveoli for lung tissue engineering (Figure 7f) [79]. In 2021, Anandakrishnan et al. proposed a fast hydrogel stereolithography printing (FLOAT) method. Through precisely controlling the photopolymerization condition, low suction force-driven, and high-velocity flow of the hydrogel prepolymer was achieved [80]. The FLOAT supports the continuous replenishment of the prepolymer solution below the curing part and the nonstop part growth. A centimeter-scale 3D printing model with hydrogels was established. It can significantly reduce the part deformation and cellular injury caused by the prolonged exposure to the environmental stresses in the layer-by-layer printing processes (Figure 7g). The multiscale embedded vascular network fabricated therein allows media perfusion needed to maintain the high cellular viability and metabolic functions in the deep core of the large-sized model.

Generally, the micro-dimensional precision of SLA printing is favorable for the construction of multiscale complex vascular structures, while taking into consideration the time-sensitive nature of live cells and tissues, the rapid printing property of this process is beneficial for the fabrication of living tissues and organs.

## 3. Bioinks That Promote Vascularization

As a basic element in bioprinting, the ideal unification of the mechanical properties and biological functions of the 3D printed constructs has always been one of the limitations of bioprinting technologies. It is reasonable that the 3D printed vascular models to have some special mechanical properties, including elasticity, tensile properties, and torsional resistance, to resist the filling pressure caused by cardiac pumping. The physiological performance needs to be similar to or higher than that of human healthy blood vessels. At the same time, due to the particularity of artificial blood vessels, bioinks are also required to have good to excellent blood compatibility, including anticoagulation. Therefore, extremely high requirements are placed on bioinks for vascular model printing, and it is also a difficulty in current research.

Bioinks that promote angiogenesis not only contain polymeric materials, cells, but also bioactive components (such as growth factors) for cell proliferation, migration, angiogenesis, and anticoagulation [95]. It has been proven that a single polymeric material cannot realize all the complex functions, so most of them are used as composite materials with or without modification. Here, we outline the developments of bioinks, which include polymeric biomaterials, cells, and bioactive factors, with the goal of promoting vascularization.

### 3.1. Biomaterials

Early in the field of vascular repair, the most frequently used materials were unbiodegradable synthetic polymers, such as polytetrafluoroethylene (ePTFE), polyethylene terephthalate (Dacron^®^) and polyurethane (PU) [96,97]. However, these materials have either poor biocompatibilities or low permeabilities for nutrient exchanges [98]. They have been seldom used in bioprinting and organ manufacturing areas.

Many biodegradable synthetic polymers, such as PU, poly(lactic acid-co-glycolic acid) (PLGA), poly (lactic acid) (PLA), poly (glycolic acid) (PGA), polycaprolactone (PCL), GelMA, hyaluronic acid methacryloyl (HAMA) and polyethylene glycol diacrylate (PEGDA), have been used in 3D bioprinting for vascular structure construction. Especially, the mechanical properties of the synthetic polymers can be regulated by adjusting the monomer or prepolymer concentration and polymerization degree. For example, the mechanical strength of photocrosslinked GelMA hydrogel can be controlled by the prepolymer concentration, functionalization degree, ultraviolet (UV) intensity, and photocrosslinking time. Compared with its precursor, the photocrosslinked GelMA hydrogel has higher porosity, viscosity and mechanical strength, which has great advantages in vascular tissue 3D bioprinting [99,100,101,102,103,104,105,106,107,108,109,110,111,112,113,114,115,116,117,118,119,120,121,122,123,124,125,126,127,128,129,130,131,132]. More importantly, most of the synthetic polymers allow compounding of bioactive factors, drugs, and specific coatings on their molecular surface with specific chemical modifications, and release them slowly into the environment afterwards.

Currently, most of the raw materials for 3D bioprinting are natural polymers, including collagen, gelatin, sodium alginate, fibrinogen, hyaluronic acid, etc. Generally, these polymers have good to excellent biocompatibilities, adjustable hardnesses and porosities, which can simulate the main components of natural ECMs. Particularly, these materials are often used as cell-laden hydrogels to print vascular constructs individually and in combination. Table 1 shows some of the commonly used synthetic and natural polymers in vascular model printing. The advantages, disadvantages and applications of these polymers in vascular model establishment have been summarized.

Especially, the polymeric hydrogels can be extruded and deposited as fiber-like structures during extrusion-based 3D bioprinting. Polymer molecules in the hydrogels resemble natural ECMs in terms of hierarchical organization and properties, providing a good microenvironment for cell adhesion, proliferation, and differentiation. Various other processing techniques, such as phase separation [133], self-assembly [134], freeze drying [135], and electrospinning [136], have been employed to fabricate fiber-like tissue engineering scaffolds. Many of the fiber-like scaffolds have been shown to be effective in promoting vascular network formation [137]. For example, Lee et al. designed a bilayered fiber-like scaffold by combining electrospinning (ELSP) and 3D printing techniques in 2019 [138]. When the fiber-like scaffold was coated with polydopamine (PDA) and VEGF was grafted directly on the PDA surface, enhanced vascular cell proliferation and angiogenic differentiation were detected.

A major problem in 3D printing of vascular networks is that the printed hollow or tubular structures are prone to collapse due to the low strength of soft and dynamic polymeric hydrogels. Over the years, researchers have explored some materials that can be removed after printing to enable the precise construction of complex hollow structures, known as “sacrificial bioinks” [139]. In this regard, the sacrificial material is shaped into the core of the desired vascular network structure and then dissolved away by physical removal, such as adjusting the pH and temperature, or chemical reaction. Several materials, such as gelatin, sugar glass, agarose, alginate, pluronic and polyvinyl alcohol, have been used as sacrificial inks, to build complex vascular networks within the 3D printed structures [68,140,141,142,143,144,145,146,147,148,149,150].

In particular, gelatin and Pluronic F127 are currently the most frequently used sacrificial biomaterials in 3D bioprinting due to their temperature-sensitive properties [139]. Emphasis should be given to the collagen-derived gelatin, which dissolves in water and becomes gelatinous at a temperature below 30 °C, and forms an aqueous solution at a temperature of 37 °C or higher, in temperature-controlled bioprinting. It can be extruded through the nozzles in a gel state and then removed at a temperature of 37 °C suitable for cell growth without any damage to the cells [151]. After 3D printing, it can be easily discarded with dissolving water.

As stated above, Hinton et al. reported a biomanufacturing method—sacrificial writing into functional tissue (SWIFT) in 2019, that introduced perfusable vascular channels in living matrices with high cellular density via embedded 3D bioprinting. These bulk SWIFTs could perfuse over long durations [152]. In their experiments, 15% gelatin was chosen as a sacrificial bioink. After the cell-laden hydrogels were patterned inside the sacrificial bioink via embedded 3D printing, the functional organization was placed in an incubator at 37 °C to remove the sacrificial bioink and yields perfusable channels in the form of single or branching conduits. The researchers then used this method to create a heart construct containing human-induced pluripotent stem cells (hiPSC) derivations, and eventually observed moderate cardiomyocyte contractility. This manufacturing method opens new avenues for constructing complex blood vessels based on 3D bioprinting.

In addition to the synthetic and natural polymers, bioactive glasses (BGs) have also been used in vascular 3D bioprinting, because some BGs have the properties to enhance angiogenesis [153,154,155,156]. Early in 2005, Day et al. showed that the conditioned medium, collected from fibroblasts and seeded with alginate beads containing 0.1 wt.% 45S5 glass particles, could increase endothelial cell proliferation rate and tubule formation phenomenon [157]. Human fibroblasts, coated with 45S5 glass particles (<5 µm), could secrete large amounts of VEGFs and basic fibroblast growth factors (bFGFs). In 2019, Jia et al. used extrusion-printed bioactive silicate and borosilicate (2B6Sr) glass scaffolds to culture MC3T3-E1 mouse pre-osteoblasts in vitro [155]. The results demonstrated that both the scaffolds were supported by VEGF assay with the capabilities for improving cells to attach, and promoting angiogenesis. At the same time, the borosilicate glass scaffold was better than silicate bioactive glass and other BGs in promoting angiogenesis at three months post-implantation. Similarly, in 2021, Zhang et al. printed a Ti-6Al-4V scaffold. After the scaffold was surface modified with mesoporous bioactive glass, it was implanted into the femoral condyles of rabbits. Bone matrix deposition and angiogenesis were then observed several weeks after implantation, indicating the enhancement of osteoconduction and vascularization during bone defect healing [158].

Additionally, some trace elements such as copper, zinc, silicon, etc., also play a role in promoting angiogenesis and are used by researchers in vascular tissue bioprinting [159,160]. The trace elements are abundant in bioceramics and BGs, so they are widely used in 3D printing of bone tissues with complex blood vessels [161,162]. For example, Fielding et al. incorporated silica (SiO_2_) and zinc oxide (ZnO) into 3D printed β-tricalcium phosphate (β-TCP) scaffold in 2013. The scaffold was then placed in the bicortical femur defects of a murine model for up to 16 weeks, and neovascularization was found to be up to three times more than the pure TCP control [163].

### 3.2. Vascularization-Directed Bioactive Substances

Bioactive substances, such as growth factors, vascular cells and anticoagulants, are often added into the polymeric hydrogels as bioinks to meet the effects of promoting angiogenesis, anticoagulation, and anti-inflammation. The reason is that simple polymers cannot achieve vascular functions close to real blood vessels with similar cell and ECM components [164]. In the following section, three major issues for vascular tissue construction are analysed.

#### 3.2.1. Growth Factor

A variety of growth factors, such as fibroblast growth factor (FGF), VEGF, transforming growth factor-β (TGF-β), have been identified as a key factor in angiogenesis [165,166,167]. For example, VEGF stimulates endothelial cell proliferation, motility, and differentiation, and VEGF-loaded hydrogels can effectively promote angiogenesis [168,169]. However, the release of growth factors via polymeric hydrogels lacks stability, and the biological activities of growth factors only lasts for a short time, so their full regenerative potential cannot always be reached. At the same time, some studies have shown that excess growth factors often increase some pathological effects [170,171,172]. High doses of VEGF may result in malformed and leaky vasculature that are obvious characteristics of tumor-site vasculatures. So, in response to these problems, additional approaches, need to be adopted to achieve slow release of growth factors, such as cross-linking them in hydrogel scaffolds by carbodiimide hydrochloride, etc. [114,173], or by encapsulating growth factors in hydrogel microspheres, through “isolation” [174,175,176,177,178,179,180].

In 2022, Rana D et al. conjugated VEGF165 to 5′-acrylate-modified aptamer and encapsulated them in GelMA hydrogels together with HUVECs and human mesenchymal stromal cells (hMSCs) for 3D culture. The release of VEGF165 was then triggered by the addition of complementary sequence (CS) based on the hybridization behavior of the 5′Alexa Fluor 488 modified CS (Fluoro-CS) with the aptamer. The control GelMA hydrogels showed high burst release of VEGF165 on day 1 (18.89 pg) and near zero release from day 2 onwards, whereas the aptamer-functionalized hydrogel exhibited minimal initial release on day 1 (3.30 pg). After the first round of triggered release with the addition of CS on day 4, the aptamer-functionalized hydrogel showed high VEGF165 compared with GelMA and the control aptamer-functionalized hydrogel (2.62 pg and 0.32 pg, respectively) at 24 released hours (20.81 pg). These results confirm the potential of acrylate–aptamer-functionalized hydrogels to sequester VEGF and control VEGF release by aptamer–CS hybridization [181].

#### 3.2.2. Heparin

Heparin is a sulfated glycosaminoglycan composed of about 32 disaccharide units with the derivatives of uronic acid and glucosamine. It is a natural anticoagulant substance in animals. Heparin-based anticoagulant and antithrombotic activity is the most widely used in vascular tissue engineering. Heparin-modified vascular implants can significantly improve blood compatibility [182,183]. However, studies have demonstrated that high doses of heparin can adversely affect endothelial cell function. Kimicata M et al. interlayered heparin-loaded GelMA hydrogels into a biomix consisting of decellularized matrix (dECM) and polypropylene fumarate (PPF) and 3D printed them through UV cross-linking. The 3D printed scaffold with 2 Sustained heparin release was achieved in the biohybrid material over a period of weeks. Endothelial function was supported with an antithrombotic environment by controlling the timing and space of heparin release [184].

Furthermore, the sulfate and carboxylate groups present in heparin molecules give it a high negative charge, which mediates its electrostatic interactions with many proteins such as growth factors, proteases, and chemokines. These proteins also have good histocompatibilities, promoting cell proliferation, growth, migration, as well as aggregate growth factors and regulating stem cell behaviors [185,186]. Researchers achieve high-density blood vessel formation in hydrogel scaffolds by combining heparin with FGF and VEGF to achieve slow control release of growth factors [187,188]. In 2016, Marchioli et al. constructed an extrahepatic islet transplantation model via 3D printing, using a 3D cyclic PCL scaffold with a heparinized surface to electrostatically bind VEGF, avoiding surrounding water for the islet encapsulation gel core to support rapid vascularization [189]. The results demonstrated that heparin fixation could increase the amount of VEGF retention up to 3.6-fold, compared to the untreated PCL scaffolds. In a chicken chorioallantoic membrane model, VEGF immobilized on the construct-enhanced angiogenesis near and on the surface of the scaffold.

### 3.3. Cells

When constructing a vascular model, polymeric biomaterials often serve as a supporting framework, bioactive factors play a guiding role for stem cell differentiation, and parenchymal cells are the key links for tissue/organ formation and maturation [190,191,192,193,194]. In another word, cells are an indispensable aspect for vascular tissue formation and maturation. This is because vascular tissues contain different cell types/ECMs and most of the adult parenchymal cells are difficult to culture in vitro for an expected period. For example, when constructing a vascularized bioartificial liver, it is necessary to incorporate parenchymal hepatocytes and mesenchymal cholangio-vascular cells (i.e., cells with structural, support, or barrier functions) to recapitulate the entire organ physiologies. During the construction of a liver vasculature, ECs are necessary for the vascular endothelium generation. Polymeric hydrogels can be used to deliver ECs. Most of the polymeric hydrogels are expected to degrade during the ECs form endothelial tissues. Many efforts are inseparable from the interactions of the co-cultured vascular ECs and other cells [50,195,196,197].

In 2013, Wang et al. developed a combined four-nozzle 3D bioprinter for complex organ manufacturing (Figure 8) [43]. The authors printed three types of cells, including hepatocytes, Schwann cells and ASCs, with a PLGA overcoat simultaneously, at specific locations with two extrusion-based nozzles and two inject-based nozzles. After 3D printing, the ASCs were induced to differentiate into ECs successfully in the 3D construct. This is the first 3D printed bioartificial liver, consisting of both branched vascular and neural networks.

In recent years, co-printing of ECs with MSCs and other adult cells have become more and more prominent for vascualr tissue/organ construction [37,198,199,200]. With the development of bioprinting technologies, most of the researchers tend to choose MSCs as the cellular components in their bioinks [201]. This is due to the multi-directional differentiation potential of stem cells, and their proliferative capabilities being much stronger than those of adult cells, which plays a key role in biomedical fields and 3D bioprinting, especially when building complex organs. The cell population is essential for vascular network formation and subsequent physiological functionalization [37,202,203,204,205,206].

In 2021, Zhou et al. printed a circular scaffold with a bioink mixed with self-assembled nanopeptides and human ASCs through an extrusion bioprinter, and carried out osteogenesis, adipogenic and endothelial induction, respectively, and successfully obtained a vascular model for directional-induced stem cell printing [207]. Recently, Skylar-Scott et al. generated patterned organoids with controlled composition and organization by simultaneously co-differentiating hiPSCs into distinct cell types through forced overexpression of transcription factors [208]. They then used this orthogonal induction of differentiation to generate ECs and neurons from hiPSCs and to generate vascularized cortical organoids.

At present, most of the allogeneic cells have been used for in vitro cultures. When the vascularized tissues/organs are printing, xenogeneic ECs, vascular SMCs and pericytes are often used to construct the vasculature. For transformational applications, autologous cells, especially MSCs and hiPSCs, are more suitable for preventing transplant rejection. Thus, stem cells have emerged as promising alternative sources of ECs and SMCs [209]. Autologous stem cells appear to be the most viable option for in vivo implantation of the 3D printed vascular tissues/organs in the future.

## 4. Challenges and Perspectives

Recently, innovations in the existing bioprinting technologies have increased rapidly, resulting in some of the structures being in a condition of “hypotaxia” to natural blood vessels. To date, most of the 3D printed vascular models find it difficult to perform the physiological functions of natural vasculatures in human organs. The 3D printed arteries, veins and/or capillaries are limited to in vitro cultures with related characterizations. It is urgently need to optimize the printing technologies and bioinks to manufacture more lifelike vascular models for in vivo implantation.

For example, by mimicking the blood flow conditions of human organs in vivo, the shear force generated by pulsatile culture in vitro can stimulate the radial alignment of ECs and activate certain signaling pathways of ECs to promote the proliferation and maturation of the next layer of SMCs. The vascular tissues can therefore exhibit a significant improvement in biomechanical properties when compared to the control static controls [210,211,212]. This needs careful consideration of the hemodynamics, anticoagulation and mechanical properties of the 3D printed bioinks for blood circulation inside. Nowadays, finite element analyses have been applied into this field for both predicting and simulating [213].

In addition, blood vessels in different organs of the human body perform specific regulatory functions, including capillaries in different organs also showing different structural characteristics. For example, heart ECs have proved much more angiogenic than other ECs derived from other organs such as liver and kidney, while liver-specific ECs have shown significantly superior supportive function in terms of albumin production from the hepatocytes [214]. The specificity of blood vessels in different organs leads to the need for more careful consideration in the construction of corresponding vascular tissues. It is reasonable to combine more cell types to the ever-evolving polymeric bioinks to mimic the key structural components of natural blood vessels as much as possible.

## 5. Conclusions

Fabrication of functional vascular networks to maintain the viability of engineered tissues is a major bottleneck in the areas of bioartificial organ engineering and manufacturing. Despite the enormous progress of 3D bioprinting technologies and bioinks in recent years, the biological complexity of vascularized tissue models poses a challenge for the development of more sophisticated platforms. Even the most state-of-the-art engineered blood vessels cannot mimic the complexities of natural vasculatures with multiscale, heterogeneous, and diverse branching structures in native organs. This requires engineers to continuously innovate new 3D printing methods. In particular, if the advantages of different printing methods can be combined and superior soft hardware can be upgraded, it will be possible to achieve the full scale complexity of vascular models for bioartificial organ engineering and manufacturing.

## Figures and Tables

**Figure 1 ijms-24-00891-f001:**
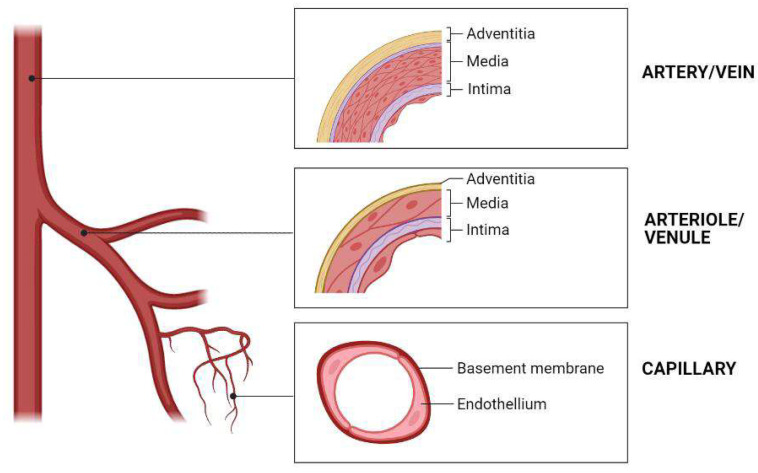
Hierarchical vascular tree and cross sections of different blood vessels, showing different blood vessel structures.

**Figure 2 ijms-24-00891-f002:**
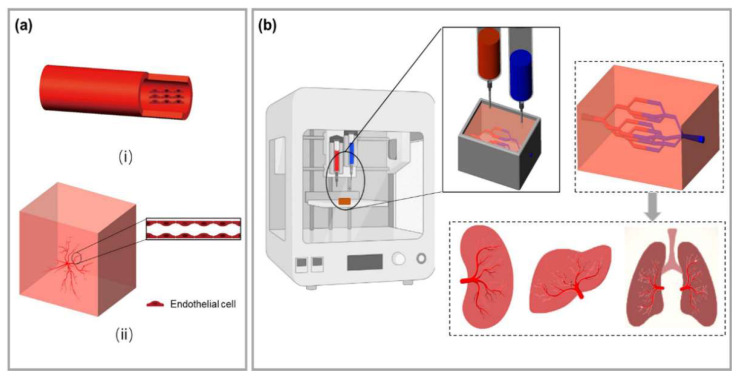
(**a**) Traditional tissue engineering blood vessel construction method. (**i**) Tubular scaffolds were prepared before endothelial cells (ECs) were seeding on the inner surface of the scaffolds; (**ii**) Stimulating capillary angiogenesis directly within the tissue engineered scaffolds. (**b**) A general depiction of 3D bioprinting for vascularized tissue construction. Multi-scale branched blood vessels can be obtained by printing ECs in a scaffold, and an ideal organoid can be obtained after in vitro cultures.

**Figure 4 ijms-24-00891-f004:**
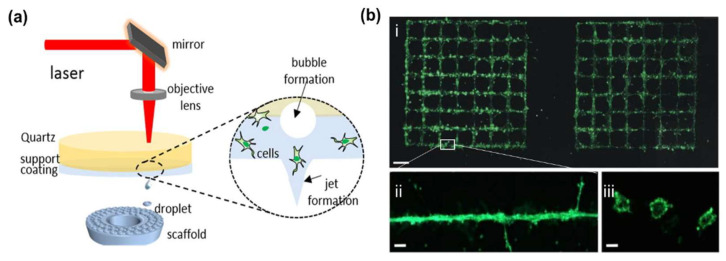
(**a**) Laser-assisted bioprinting based on laser-induced forward transfer. Adapted from Ref. [33]. (**b**) Green fluorescent labeled endothelial cells printed in the patterns (**i**) of grids on Matrigel, (**ii**) the enlarged line of (i), (**iii**) cross-sections of the printed vascular structures. Adapted from Ref. [34].

**Figure 6 ijms-24-00891-f006:**
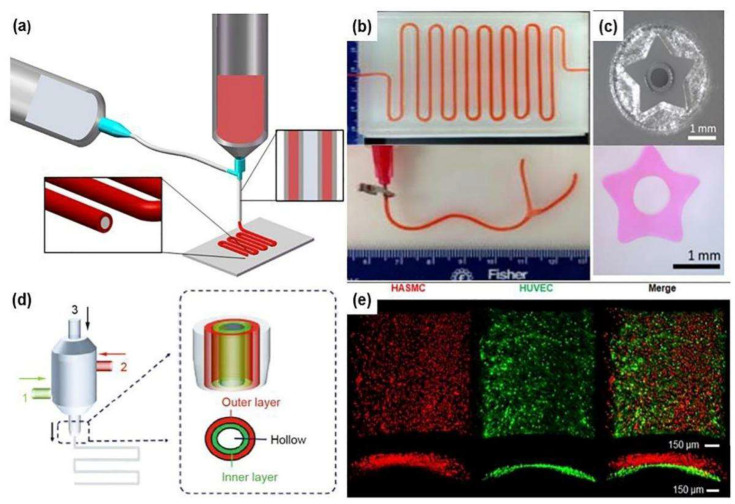
(**a**) Schematic illustration of a coaxial bioprinting technique. (**b**) A long vasculature conduit printed in a zigzag pattern with perfused cell culture media, and a branched vasculature, reported by Zhang et al. Reprinted from Ref. [67]. (**c**) A customized nozzle with a 5-point star geometry and the cross-section of the 5-point star tube produced using the nozzle, reported by Millik et al. Reprinted from Ref. [68]. (**d**) Schematic illustration of a multichannel coaxial extrusion system, reported by Pi et al. Two cellular bioinks are delivered respectively through the openings 1, 2 to form the second (i.e., middle) and third (i.e., outer) layers of the vessels, while a sacrificial material is delivered through the opening 3 to form the inner layer of the vessel. Reprinted from Ref. [70]. (**e**) A double-layered blood-vessel-like structure produced using a triple concentric nozzle, in which, human smooth muscle cells (hSMCs) in red and human umbilical vein endothelial cells (HUVECs) in green at day 5, reported by Bosch et al. Reprinted from Ref. [72].

**Figure 7 ijms-24-00891-f007:**
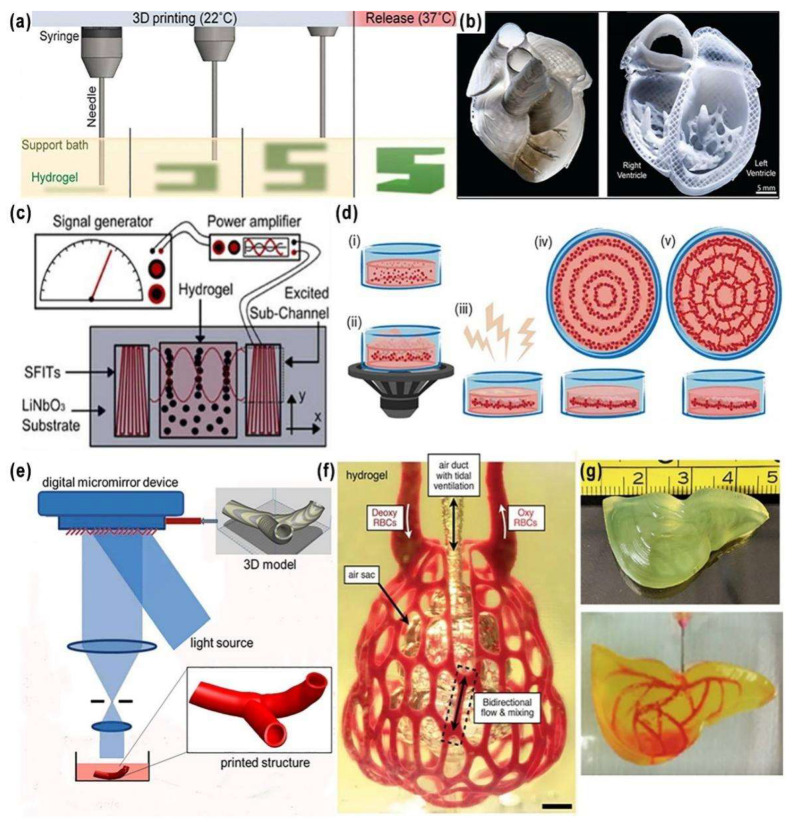
(**a**) A schematic of the FRESH process showing the hydrogel (**green**) being extruded and cross-linked within the gelatin slurry support bath (**yellow**). The 3D object is built layer-by-layer and, when completed, is released by heating to 37 °C and melting the gelatin, reported by Hinton et al. Reprinted from Ref. [75]. (**b**) A FRESH-printed collagen heart (**left**), and cross-sectional view of the collagen heart (**right**), showing the left and right ventricles and interior structures, reported by Lee et al. Reprinted with permission from Ref. [76]. Copyright © 2023, The American Association for the Advancement of Science. (**c**) Schematic description of the acoustic printing device, based on surface acoustic waves (SAW), reported by Naseer et al. Reprinted from Ref. [77]. (**d**) Sound-induced morphogenesis (SIM) based on Faraday waves, (**i**–**v**) are schematic diagrams of the formation of vascular network, reported by Petta et al. Reprinted from Ref. [78]. (**e**) Stereolithography bioprinting of branch blood vessels. (**f**) A vascularized alveolar model topology, reported by Grigoryan et al. Reprinted from Ref. [79]. (**g**) A liver model with smooth surface and monolithic, translucent hydrogel body, in which, a vascular channel network was filled, stained with rhodamine B and visualized under fluorescence, reported by Anandakrishnan et al. Adapted from Ref. [80].

**Figure 8 ijms-24-00891-f008:**
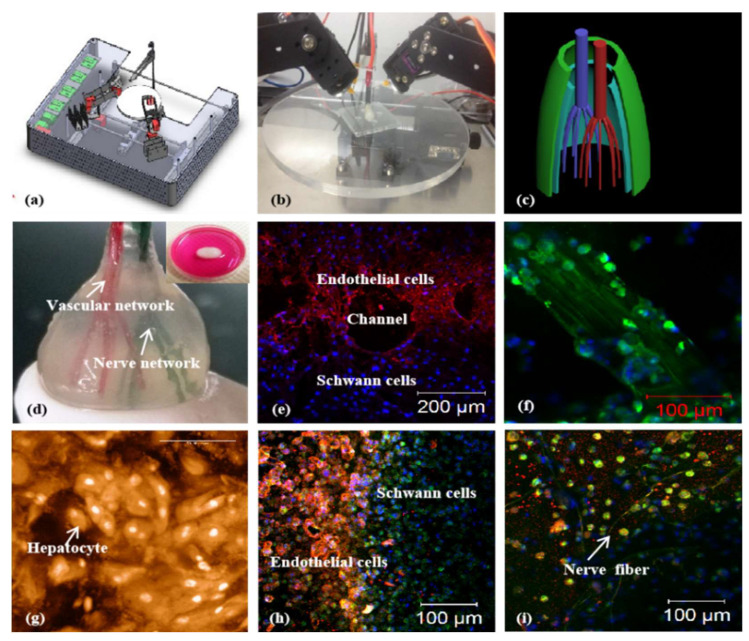
A combined four-nozzle 3D organ bioprinting technology created in Prof. Wang’s laboratory at Tsinghua University in 2013. Adapted from Ref. [49]: (**a**) equipment of the combined four-nozzle 3D organ bioprinter; (**b**) working state of the combined four-nozzle 3D organ printer; (**c**) a CAD model representing a large-scale vascularized and innervated hepatic tissue; (**d**) a semielliptical 3D construct containing a poly (lactic-co-glycolic acid) (PLGA) overcoat, a hepatic tissue made from hepatocytes in a gelatin/chitosan hydrogel, a branched vascular network with fully confluent endothelialized ASCs on the inner surface of the gelatin/alginate/fibrin hydrogel, and a hierarchical neural network made from Schwann cells in the gelatin/hyaluronate hydrogel; the maximal diameter of the semiellipse can be adjusted from 1 mm to 2 cm according to the CAD model; (**e**) a cross section of (**d**), showing the endothelialized ASCs and Schwann cells around a branched channel; (**f**) a large bundle of nerve fibers formed in (**d**); (**g**) hepatocytes underneath the PLGA overcoat; (**h**) an interface between the endothelialized ASCs and Schwann cells in (**d**); (**i**) some thin nerve fibers.

**Table 1 ijms-24-00891-t001:** The commonly used polymers in vascular model printing.

Biomaterial	Advantage	Deficiency	Application	Reference
Gelatin	Excellent biocompatibility, good cell adhesion, physical crosslinking properties	Low shape fidelity, especially unstable at temperatures suitable for cell growth, and low mechanical strength	Modification such as methacryloyl anhydride, or cross-linking, enhances its mechanical strength and printing resolution	[26,50,99,100,101,102,103,104,105]
PU	Excellent histocompatibility, super mechanical strength	Cells cannot be encapsulated directly	3D printing vascular networks, bioartificial liver manufacturing	[106,107,108,109,110,111]
PLGA	Poor biocompatibility, middle mechanical properties	Cells cannot be encapsulated directly	3D printing vascular networks, bioartificial liver manufacturing	[112,113,114,115]
Alginate	Shear thinning properties, very short time polymerizable, porous properties	Poor biocompatibility, low cell adhesion properties	Often mixed with gelatin, hyaluronic acid, etc. for printing; as a sacrificial material for vascular stents	[47,53,116,117]
Fibrinogen	Excellent biocompatibility, good cell adhesion	Low mechanical strength, fast degradation rate	Commonly used for thrombin cross-linking, blending or double cross-linking with gelatin, sodium alginate, etc.	[118,119]
Hyaluronic Acid	High water absorption, excellent biocompatibility, low molecular weight have the ability to promote cell proliferation	Low mechanical strength and poor formability	Modification such as methacryloyl anhydride, or compounded with other materials	[101,120,121,122]
dECM	Promotes cell adhesion, proliferation and functionalization, especially has a certain antithrombotic effect	Low mechanical strength, slow gelation, complicated preparation process	Often used with fast cross-linking materials such as sodium alginate	[123,124,125,126,127]
Pluronic ^®^ F127	High resolution printing, special temperature sensitive properties	Low mechanical strength, fast degradation rate	As a sacrificial material for vascular stents	[26,63,68,128]

## Data Availability

Not applicable.

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
