# Peer review of "Bioprinting Technologies and Bioinks for Vascular Model Establishment"

_ijms, 2023, doi:10.3390/ijms24010891_

Round 1
Reviewer 1 Report
The review article discusses the various techniques used in vascular 3D bioprinting and materials which can be used as bioink for such applications. Supporting figures have been incorporated from multiple sources. Below are the comments.
1. Citation needs thorough checking. (specific corrections have been included in the attached file.
2. Year of citation is missing in most of the places in the text.
3. Sentences are long and lack clarity.
4. Repetition of text was found in section 2.2
5. Figure ligands are interchanged for figure 3 b & c
6. Sources of figures and required permissions should be mentioned clearly for all the figures.
7. The text needs thorough checking for spelling mistakes and sentence reframing. It has been marked for a few in the attached file.
The article provide comprehensive information about the 3D bioprinting of the vascular network and will be helpful for the scientific community working in the area
Author Response
Dear Mr. Popovici:
Thank you for processing our manuscript and the reviewers’ comments concerning our manuscript entitled “Bioprinting Technologies and Bioinks for Vascular Model Establishment” (ID: ijms-2072795)!
We appreciate the time and effort that you and the reviewers have dedicated on our manuscript and are grateful for the insightful comments on and valuable improvements to our paper.
We have studied the comments carefully and made corrections which we hope can be appra approved. All the revised portions are marked in red throughout the manuscipt. The main corrections in the paper and the responds to the reviewer’s comments are as flowings:
Reviewer #1:
- Response to comment: (Citation needs thorough checking.)
Response: We are very sorry for the negligence of citation format. We have modified the reference format and checked it for correctness.
- Response to comment: (Year of citation is missing in most of the places in the text.)
Response: We have added the year of citation where it is cited in the text.
- Response to comment: (Sentences are long and lack clarity.)
Response: We apologize for the poor sentence expression. We have worked on the manuscript for a long time and revised it carefully. We have now worked on both language and readability. We really hope that the language level has been improved substantially.
- Response to comment: (Repetition of text was found in section 2.2)
Response: We feel sorry for our carelessness. In the resubmitted manuscript, we have removed the repetitive content in section 2.2.
- Response to comment: (Figure ligands are interchanged for figure 3 b & c.)
Response: We have replaced the position of (b) and (c) in Figure 3, which makes our representation more reasonable.
- Response to comment: (Sources of figures and required permissions should be mentioned clearly for all the figures.)
Response: We have illustrated the sources of figures. For citations without open access, we have obtained the necessary permission from the copyright holder.
- Response to comment: (The text needs thorough checking for spelling mistakes and sentence reframing. It has been marked for a few in the attached file.)
Response: The text have been revised totally in red color.
Best wishes and regards,
Xiaohong Wang
Author Response
Reviewer #2:
- Response to comment: (Templated cell-based technique and its application were well-described.)
Response: We sincerely thank the reviewer for his/her careful reading.
- Response to comment: (2.3. D bioprinting technology for building blood vessels needs to be changed to “2. 3D bio....”)
Response: we have made changes to this issue.
- Response to comment: (Page 4, first paragraph: need to remove repeating word “that is related Related”)
Response: The repetitive word in Page 4 has been removed.
- Response to comment: (Page 4, last paragraph: microvasc0ular=> microvascular)
Response: In the resubmitted manuscript, the typo has been revised.
- Response to comment: (Format needs to be changed of section 2.3)
Response: Section 2.3 has been modified according to the normative format.
- Response to comment: (In LAB, most of the added literatures were awesome but older, this reviewer suggests to explore the recent advancement of LAB toward vasculature and add them in Lab section.)
Response: We sincerely appreciate the valuable comments. We carefully reviewed the relevant literature and reports on the application of LAB toward vasculature in the last three years. In the supplement, we introduced a new LAP printing technique, called Laser Induced Side Transfer (LIST), which is more effective than traditional methods in vascularized printing. We have added this content in the first paragraph of page 6.
- Response to comment: (Importance of short bio-fiber in effort to regenerate vasculature can a section or an added paragraph.)
Response: Based on the reviewer's comments and the relevance of our paper, we have added a paragraph in section 2.4 (page 14) to describe the preparation of fibrous materials and their effective applications in inducing angiogenesis.
- Response to comment: (This reviewer thought contribution of extrusion-based bioprinting could be extended adding some recent innovative efforts. Please add some recent works in these sections 2.4 and 2.5.)
Response: We have added some recent works in section 2.4 and section 2.5 according to the Reviewer’s suggestion. For example, in section 2.3, we have added a method for cellular self-assembly combined with extrusion-based printing that enables vascular construction from the centimeter scale to the capillary scale within printed scaffolds. This is certainly a good exploration for extrusion printing. We have added two paragraphs at the end of this section, on page 7. In section 2.4, we have added the exploratory application of coaxially printed tubular stents with in vivo vascular anastomosis. We have added this content in Page 9.
Additional supplement: We have tried our best to revise our manuscript according to the comments. For the sake of content completeness, we have added a case study of stem cell directed induced differentiation in a printed scaffold in section 3.3, with figure 8 supporting the relevant content, on page 14.
We would like to express our great appreciation to the editors and reviewers for our manuscript and are looking forward to hearing from you soon.
Round 2
Reviewer 1 Report
The authors have answered all the queries and corrected as suggested. Article in the present form is acceptable, however, spell check is required as still there are some spelling mistakes.
For eg esection 2 line 5 dvantages is written
Section 2.2 it is typed as Ttransfer (LIST) [38].
such mictakes should be corrected. Thi scan be correcetd as prood reading step